# Clonal Diversity of *Candida auris*, *Candida blankii*, and *Kodamaea ohmeri* Isolated from Septicemia and Otomycosis in Bangladesh as Determined by Multilocus Sequence Typing

**DOI:** 10.3390/jof9060658

**Published:** 2023-06-12

**Authors:** Fardousi Akter Sathi, Meiji Soe Aung, Shyamal Kumar Paul, Syeda Anjuman Nasreen, Nazia Haque, Sangjukta Roy, Salma Ahmed, Mohammad Monirul Alam, Shahed Khan, Mohammad Arif Rabbany, Joy Prokas Biswas, Nobumichi Kobayashi

**Affiliations:** 1Department of Microbiology, Mymensingh Medical College, Mymensingh 2200, Bangladesh; drsathi.dmc@gmail.com (F.A.S.); nasreenm19@gmail.com (S.A.N.); drnaziahaque@gmail.com (N.H.); drsangjukta@gmail.com (S.R.); 2Department of Hygiene, School of Medicine, Sapporo Medical University, Sapporo 060-8556, Japan; meijisoeaung@sapmed.ac.jp; 3Netrokona Medical College, Netrokona 2400, Bangladesh; drshyamal10@yahoo.com; 4Mugda Medical College, Dhaka 1214, Bangladesh; ahmed.salma51@yahoo.com; 5Department of ENT, Mymensingh Medical College Hospital, Mymensingh 2200, Bangladesh; monirul.dmc@gmail.com; 6Department of Oral Microbiology, Mymensingh Medical College Hospital, Mymensingh 2200, Bangladesh; shahed.khan1988@gmail.com; 7Department of Neonatology, Mymensingh Medical College Hospital, Mymensingh 2200, Bangladesh; arifdrmmc@gmail.com; 8Department of Pathology, Netrokona Medical College, Netrokona 2400, Bangladesh; joyprokasbiswas@gmail.com

**Keywords:** *Candida auris*, *Candida blankii*, *Kodamaea ohmeri*, multilocus sequence typing (MLST), ST, Bangladesh

## Abstract

*Candida auris*, *Candida blankii*, and *Kodamaea ohmeri* have been regarded as emerging fungal pathogens that can cause infections with high mortality. For genotyping of *C. auris*, a multilocus sequence typing (MLST) scheme based on four locus sequences has been reported, while there is no typing scheme for *C. blankii* and *K. ohmeri*. In the present study, the existing MLST scheme of *C. auris* was modified by adding more locus types deduced from sequence data available in the GenBank database. Furthermore, MLST schemes of *C. blankii* and *K. ohmeri* were developed using the four cognate loci (ITS, *RPB1*, *RPB2*, D1/D2) and similar sequence regions to those of *C. auris*. These MLST schemes were applied to identify the ST (sequence type) of clinical isolates of *C. auris* (*n* = 7), *C. blankii* (*n* = 9), and *K. ohmeri* (*n* = 6), derived from septicemia or otomycosis in Bangladesh in 2021. All the *C. auris* isolates were classified into a single ST (ST5) and clade I, having a Y132F substitution in ERG11p, which is associated with azole resistance. Similarly, all the *C. blankii* isolates belonged to a single type (ST1). In contrast, six *K. ohmeri* isolates were assigned to five types (ST1-ST5), suggesting its higher genetic diversity. These findings revealed the availability of MLST schemes for these three fungal species for understanding their clonal diversity among clinical isolates.

## 1. Introduction

*Candida* species are the most common fungal pathogens in humans, causing superficial and systemic infections. Among the forms of candidiasis, candidemia is among the leading causes of nosocomial infections, associated with significant mortality, and its incidence has been increasing worldwide in the past few decades [1]. A notable recent trend in epidemiology of *Candida* is the emergence of novel *Candida* species such as *C. auris* [2], and the prevalence of uncommon pathogenic yeast [3], which has been suggested to be due mostly to an increase in immunocompromised patients.

*Candida auris* is a newly described species that was first isolated from the external ear canal of a patient in Japan in 2009 [4]. Subsequently, the global spread of this species has occurred rapidly [5,6]. However, it has also been noted that the genetically different clades (I through V) of *C. auris* emerged nearly simultaneously in multiple countries across six continents, for an unknown reason [7]. Bloodstream infection is the most common type of infection by *C. auris* in children, with a high mortality rate [6]. This species shows high resistance to fluconazole, with variable susceptibility to other antifungals and an increase in multi-drug resistance (MDR) [5,6]. The spread of *C. auris* with MDR has been recognized as a public health concern in Europe [8] and the Middle East [9], and its sudden increase in prevalence in 2021 was reported in the US [10]. 

*Candida blankii*, which was first described in mink in 1968 [11], has been isolated from dairy products. However, since its identification in a neonate with cystic fibrosis in 2015 [12], *C. blankii* has been considered an emerging human pathogen that primarily infects immunocompromised humans, mostly causing bloodstream infections [3,13,14]. Reduced susceptibility to azoles in this species was reported [15], and an outbreak of fungemia was described in India [16]. Recently, *C. blankii* bloodstream infection in the elderly was reported as a complication of COVID-19 in the US [17].

*Kodamaea ohmeri*, a heterotypic synonym of *C. guilliermondii var. membranaefaciens*, is also regarded as an emerging fungal pathogen, primarily causing fungemia, endocarditis, and onychomycosis [18,19]. Although amphotericin B and fluconazole are commonly used as antifungal therapy, the occurrence of fluconazole resistance is noted [19,20]. Mortality due to *K. ohmeri* infection is high in cases of fungemia, but low in other infection types. Infections with this species have been found worldwide, with nearly 70% being reported in Asia (particularly East and Southeast Asia) [19].

In Bangladesh, candidemia is estimated to constitute a serious burden, with an estimated 8100 cases per year (5/100,000 rate) [21]. We previously characterized the species and antimicrobial susceptibility of 109 presumptive *Candida* isolates from 3 infection types in Mymensingh, Bangladesh, and revealed 13 species, including *C. auris* and *K. ohmeri* [22]. We describe herein the detection of more isolates of *C. auris* and *K. ohmeri*, and the identification of *C. blankii,* firstly in Bangladesh, among isolates at the same study site. Furthermore, to clarify the clonal diversity, isolates of the three emerging fungal species were analyzed using multilocus sequence typing (MLST), based on partial nucleotide sequences of four housekeeping genes.

An MLST scheme of *C. auris* using four loci (ITS, *RPB1*, *RPB2*, and D1/D2) of housekeeping genes was originally proposed by Cendejas-Bueno and coworkers [23]. Using these sequences of the four loci, Prakash and coworkers reported a multilocus phylogenetic analysis of *C. auris* [24]. Subsequently, employing the sequence data of the same four loci, Kwon et al. [25] designed an MLST scheme in which the locus type was assigned to the sequences, and the sequence type (ST) was assigned to the combination of the four locus types (allelic profiles). This scheme was utilized for 61 Korean isolates, revealing a single ST (ST cluster) among them. However, this MLST analysis for *C. auris* has not yet been well utilized, and has not been reported for more clinical isolates at other study sites. Furthermore, no MLST scheme has been established for *C. blankii* and *K. ohmeri*. In the present study, the MLST scheme of *C. auris* was improved through the addition of more types of four loci, the sequences of which could be retrieved from the GenBank database. For *C. blankii* and *K. ohmeri*, provisional MLST schemes were created, based on the four loci, which are same as those of *C. auris*, using the sequence data retrieved from the GenBank database. These MLST schemes were applied for genotyping recent clinical isolates of *C. auris*, *C. blankii* and *K. ohmeri* in Bangladesh.

## 2. Materials and Methods

### 2.1. Clinical Isolates of C. auris, C. blankii, and K. ohmeri

In our previous study, clinical isolates of presumptive *Candida spp.* collected from Mymensingh Medical College hospital, during 10-month period in 2021, were characterized [22]. In the present study, we analyzed additional 22 isolates that were collected in the same period but could not be identified phenotypically, and by the PCR-RFLP method for the ITS region. These isolates were recovered from the blood and aural swabs (discharge) of patients with suspected candidemia and otomycosis, respectively. Isolation of *Candida* from specimens was performed as described previously [22].

The nucleotide sequence of the ITS region was determined using Sanger sequencing, with PCR products and *Candida* species universal primers ITS1 and ITS4 [26]. Species were determined by searching for an identical or highly similar sequence in the GenBank database, using the BLAST web tool (https://blast.ncbi.nlm.nih.gov/Blast.cgi, accessed on 20 April 2023). Isolates identified as *C. auris*, *C. blankii*, and *K. ohmeri* were further genotyped using MLST schemes.

### 2.2. Antifungal Susceptibility Testing

Susceptibility to antifungal agents was determined using the broth microdilution method, as described previously [22]. The minimum inhibitory concentration (MIC) was determined for fluconazole and amphotericin B. The susceptibility of *C. auris* to these antifungals was judged according to the criteria defined by the US Centers for Disease Prevention and Control (CDC) (https://www.cdc.gov/fungal/candida-auris/c-auris-antifungal.html, accessed on 26 May 2023).

### 2.3. MLST Schemes of C. auris, C. blankii and K. ohmeri

MLST was performed according to the procedure described previously by Kwon et al. [25]. In this scheme, partial nucleotide sequences are determined for four loci: the ITS region of the ribosomal subunit, the D1/D2 large ribosomal subunit region, *RPB1* encoding the largest subunit of RNA polymerase II, and *RPB2* encoding the DNA-dependent RNA polymerase II. Fungal DNA was extracted using the phenol-chloroform method, and the ITS, D1/D2 regions and *RPB1* sequences (i.e., the same sequence regions among the three species) were amplified by PCR, using the primers and conditions described previously [24] (Table 1). For PCR amplification of *RPB2*, newly designed primers were used for the three species (Table 1). Nucleotide sequences of amplified PCR products were determined via Sanger sequencing on an automated sequencer.

For *C. auris*, obtained sequence data were collated with those published previously [25] to assign type of each locus. Furthermore, for all the three fungal species, sequences representing different types of the four loci were searched using the BLAST web tool, utilizing the obtained sequence data for the present study isolates. The sequence data of individual allelic types were compared through alignment using the Clustal Omega program (https://www.ebi.ac.uk/Tools/msa/clustalo/, accessed on 10 April 2023). Phylogenetic dendrograms of the sequences of four MLST loci were constructed using the maximum likelihood method and MEGA.6 software, together with sequence data available in the GenBank database. ST (sequence type) was assigned to the combination of the four locus types, i.e., the allelic profile.

### 2.4. Sequence Analysis of ERG11 Gene of C. auris and Clade Typing

The nucleotide sequence of *C. auris ERG11* gene was determined as described previously [27]. To detect mutation associated with azole resistance, deduced amino acid sequences were compared with those of the fluconazole-susceptible *C. auris* strain B11220 (a wild-type strain of clade II, GenBank accession no. XP_028891800) [28]. The major geographical clade of *C. auris* (I-IV) was determined using clade-specific PCR, as described previously [29].

### 2.5. GenBank Accession Number

The nucleotide sequences of ITS and D1/D2 regions, *RPB1* and *RPB2* determined in the present study were deposited in the GenBank database under the accession numbers shown in Appendix A.

## 3. Results

### 3.1. Identification of Fungal Species

The 22 isolates were identified as *C. auris* (*n* = 5), *C. blankii* (*n* = 9), *C. parapsilosis* (*n* = 2), and *K. ohmeri* (*n* = 6), using sequencing analysis of the ITS region. Among them, *C. auris*, *C. blankii*, and *K. ohmeri* isolates (*n* = 20) were further analyzed genetically, along with two *C. auris* isolates that had been reported previously [22]. These total 22 isolates were derived from septicemia (*n* = 18: 7 *C. auris*, 8 *C. blankii*, and 3 *K. ohmeri*) or otomycosis (*n* = 4; 1 *C. blankii* and 3 *K. ohmeri*).

### 3.2. MLST Scheme of C. auris

The nucleotide sequences of the four MLST loci were determined for *C. auris* isolates, and the sequence ranges were adjusted according to those reported previously [25]. Sequences of ITS, *RPB1*, D1/D2 of all the seven isolates were identical to those of allelic types “b”, “b”, “b” designated by Kwon et al. [25], respectively. The *RPB2* sequence was identical among the seven isolates, but did not coincide with those of the alleles “a” or “c”, as described previously [25]. In the present study, the allelic designations using the alphabet (a–c) were converted to Arabic numerals, because the allelic profiles of all the established MLST schemes of the bacterial species and four *Candida* species are shown as combinations of Arabic numerals (https://pubmlst.org/, accessed on 10 April 2023). Accordingly, four alleles (ITS-*RPB1*-*RPB2*-D1/D2) of the present isolates were assigned to 1-1-1-1 (Table 2), and the sequences of the four loci are shown in Appendix A. To assign more allelic types for the *C. auris* MLST scheme, divergent sequences of the four loci were searched among the GenBank database using the BLAST web tool and the allelic type 1 sequence of the four loci as standards. Allelic type numbers were assigned according to higher frequency, as detected in the GenBank database. Thus, the allelic types of four loci representing divergent sequences were provisionally assigned (Table 2 and Appendix A). Although less than four alleles were found in D1/D2, *RPB1*, and *RPB2*, a larger number of divergent sequences were retrieved for ITS. Therefore, in this study, allelic types were assigned to only six sequences of ITS that showed a frequency of two or more in the GenBank database (Table 2). Sequence identities within ITS and D1/D2 loci were somewhat lower (95.19–99.74% and 96.46–99.26%, respectively) than those within *RPB1* and *RPB2* (≥98.52%) (Appendix A). The allelic types of four loci were evidently distinguished by phylogenetic dendrograms (Figure 1).

### 3.3. MLST Schemes of C. blankii and K. ohmeri

MLST schemes for *C. blankii* and *K. ohmeri* were designed using a range of sequences in the four loci which are similar to those of *C. auris* scheme [25] (Appendix A). As described above, the divergent sequences of the four loci were searched using the BLAST web tool and the sequences obtained in the present isolates; allelic types were assigned as shown in Table 2, and also schematically represented by dendrograms (ITS locus of *K. blankii* and four loci of *K. ohmeri*) in Figure 1. For *C. blankii*, four alleles were identified for ITS, while only one or two alleles were assigned to other loci. In contrast, eight ITS and *RPB2* allelic types were found in *K. ohmeri*, while the ITS region was more divergent (94.52–99.74% identity) than *RPB2* (98.69–99.8% identity) (Appendix A).

### 3.4. Assignment of ST for C. auris, C. blankii, and K. ohmeri Isolates

To assign ST for isolates of individual species, allelic profiles representing a combination of the four loci (ITS-*RPB1*-*RPB2*-D1/D2) were summarized in Table 3. All the *C. auris* isolates in the present study had a 1-1-1-1 profile, which was assigned to ST5, following ST1 through ST4 (ST cluster), which had been already described [25]. Nine *C. blankii* isolates exhibited an identical profile of 1-1-2-1, which was indicated by ST1. Six *K. ohmeri* isolates showed five different allelic profiles, representing ST1 through ST5, due to more diverse allelic types in RPB1 and RPB2. Only two isolates from septicemia belonged to the same type, ST1 (Table 4).

### 3.5. Characteristics of C. auris, C. blankii, and K. ohmeri and Their Infections

All the *C. auris* isolates in the present study were classified into clade I (the South Asia clade), and mostly showed resistance to fluconazole and amphotericin B (Table 4). In the ERG11p, all isolates had a Y132F substitution, without a K143R mutation. Except for a case of otomycosis with *C. blankii*, all the *C. auris* and *C. blankii* infections were septicemia that occurred in neonates and was often associated with premature birth, a low birth weight, and an intravenous canula as risk factors. Most septicemia cases with *C. auris* (6/7), and half of septicemia cases with *C. blankii* (4/8) were fatal. Two *C. blankii* isolates showed higher MIC to fluconazole and amphotericin B than other isolates. *K. ohmeri* infections were septicemia in neonates (*n* = 3) and otomycosis in adults (*n* = 3), without a fatal case. Otomycosis due to *K. ohmeri* was recurrent and associated with chronic suppurative otitis media. Three isolates from otomycosis showed higher MIC to fluconazole than blood isolates. The two infection types were caused by different STs of *K. ohmeri*.

## 4. Discussion

In our previous study, *C. auris* and *K. ohmeri* were genetically confirmed in a few among 109 isolates of presumptive *Candida* species in Bangladesh [22]. However, in the present study, further characterization of those remaining among the same batch of isolates revealed the presence of more isolates of these species, and *C. blankii* was first confirmed in Bangladesh. Consequently, *C. auris*, *C. blankii*, and *K. ohmeri* were identified at a rate of approximately 6% each (8–9 among a total 131 isolates), accounting for nearly 20% of the total *Candida*-like yeast isolates. Particularly, *C. auris* and *C. blankii* were isolated from fatal septicemia, and *K. ohmeri* was associated with recurrent otomycosis as well as neonatal septicemia. These findings underscore the importance of identification of the three emerging species as fungal pathogens in Bangladesh. However, these species might have been underestimated or not recognized, because they are easily misidentified as other *Candida* species using biochemical testing [2,5,16,18]. Therefore, sequencing analysis of the ITS region should be employed for notable cases, both for appropriate therapeutic management and epidemiological study.

*C. auris* isolates in the present study were revealed to be a single clone belonging to ST5 (locus profile, 1-1-1-1) and clade I (South Asia clade), having an identical substitution in ERG11p (Y132F). The similar single clonality of *C. auris* was reported in Korea for 61 isolates from different sources and hospitals over a 10-year period [25]. These isolates belonged to ST2 (locus profile, 2-2-2-2) and were grouped into clade II (East Asia clade), and associated with a low incidence of the ERG11p substitution. Taken together, these findings using MLST indicated the distinctive nature of *C. auris* in global regions (represented by clades), and the single clonality of isolates at each study site. The distinction of isolates from South Asia and East Asia was also described using a phylogenetic analysis of the *C. auris* gene sequence (a concatenated sequence of four MLST loci), revealing cluster 1 (India, South Africa), cluster 2 (Korea and Japan), and cluster 3 (Brazil) [24]. Similarly, MLST-based phylogenetic analysis showed the differentiation of clades I-III among global isolates [30], and the distinction between South African isolates and Korean isolates [31].

Kwon and coworkers also analyzed ten *C. auris* strains representing each of four clades, provided by the US CDC, using the MLST scheme, and showed that ST1, ST2, ST3, and ST4 correspond to clade I, II, III, and IV, respectively [25]. However, in our present study, all the *C. auris* isolates in Bangladesh, which belonged to clade I, were assigned to ST5. This difference in the ST of clade I may imply a diversity of clones within a clade. In addition, the ST1 CDC isolate AR0382 had no mutation in Erg11p [25], unlike our ST5 isolates. Therefore, it is suggested that the *C. auris* strains of clade I may have evolved, presumably through exposure to antifungal agents, while a single *C. auris* clone is distributed throughout a region/country.

In the present study, in Bangladesh, *C. blankii* was first identified, and the isolates were revealed to comprise a single clone (ST1), causing 50% of fatal cases among septicaemia. Although MLST was not available previously, a single clonal origin of *C. blankii* isolates was described in a study in Delhi, India, as revealed by an amplified fragment length polymorphism pattern [16]. These isolates were derived from fungemia in neonates (with a case fatality rate of 45%), showing similar observations to our study, which suggests a prevalence of *C. blankii* in South Asia. Although limited information is available on this subject, human infections due to *C. blankii* have been reported in South Asia, Brazil, Argentina, Kuwait, Iran, and India [13,14,15,16,17,32,33]. Therefore, South Asia appears to be one of the global hotspots of the prevalence of *C. blankii*. Indian *C. blankii* isolates, which were collected mostly in 2016, were all susceptible to fluconazole [16]. However, in our study, two among nine *C. blankii* isolates (22%) showed higher MICs to fluconazole and amphotericin B, causing fatal outcomes. Additionally, in Kuwait, fluconazole-resistant *C. blankii* was reported to cause a fatal bloodstream infection in a neonate [13]. These findings suggest the occurrence of a mutation associated with azole resistance in *C. blankii,* as observed for *C. auris*. Therefore, the prevalence and spread of azole resistance in *C. blankii* should be also carefully monitored.

In contrast to *C. auris* and *C. blankii*, the *K. ohmeri* isolated in our present study was genetically divergent, as defined by ST, despite the small number of isolates. The isolates were derived from septicemia in children and otomycosis in adults, with the latter showing higher MICs to fluconazole. This species has been isolated from patients with various types of infections and background, with fungemia in neonates and elderly being dominant [18,19]. Although molecular epidemiological information is not available for *K. ohmeri* isolates, it is suggested that the wide range of infection types may be related to potentially higher genomic diversity, which might allow adaptation to various human body sites and tissues. This point may be further understood using genotypic analysis of *K. ohmeri,* depending on the infection type and/or host attributes. ST1 was assigned to two isolates from bloodstream infection in a neonate and in an infant, which may suggest an association of this type with septicemia. Though five STs were identified for the present *K. ohmeri* isolates, the D1/D2 locus was only assigned to a single type (type 1). This may imply that this locus type is the one prevalent in Bangladesh, or has higher conservation in D1/D2 than other loci within this species.

To date, MLST schemes of *Candida* have been established for four species (*C. albicans*, *C. glabrata*, *C. krusei*, and *C. tropicalis*), employing seven loci (*C. albicans*) or six loci (other species) [34,35,36,37]. According to the PubMLST website (https://pubmlst.org/organisms/, accessed on 29 April 2023), >4000 types of *C. albicans*, and >1400 types of *C. tropicalis* were registered, but there are still <300 types for *C. krusei* and *C. glabrata*. In the present study, we modified the existing MLST scheme of *C. auris* [25], and designed new MLST schemes of *C. blankii* and *K. ohmeri* based on the *C. auris* scheme with four loci. With these MLST schemes, we identified only a single ST for *C. auris* and *C. blankii*, and five STs for *K. ohmeri*. The lesser variety in identified STs in the present study may be due to the limited numbers of clinical isolates available for the analysis, and also possibly to the less divergent nature of these species within a single study site. However, there are many sequences representing various locus types of the four loci that could be retrieved from GenBank, and many of these types were not assigned to the present isolates in Bangladesh. Therefore, the presence of numerous STs may actually be estimated for global isolates. To evaluate the discrimination ability and usefulness of the MLST schemes for the epidemiological study of *C. auris*, *C. blankii*, and *K. ohmeri*, the accumulation of more sequence data from clinical isolates will be necessary. Above all, because the sequence data of *C. blankii* are substantially lacking, it is essential to have medical professionals recognize the presence of this emerging species to facilitate its genetic identification in clinical laboratories. For this, the spread of accurate identification of *Candida* species using sequencing of ITS region is necessary.

A limitation of the present study is that short tandem repeat (STR) typing was not performed for *C. auris* [38] because we focused on the design and application of MLST schemes for the three fungal species. STR typing has been applied for the genotyping of *C. auris* as a rapid and reliable method with high discrimination ability [39]. Therefore, a comparison of MLST with STR would be also of significance for the development of the molecular epidemiology of *C. auris*.

## Figures and Tables

**Figure 1 jof-09-00658-f001:**
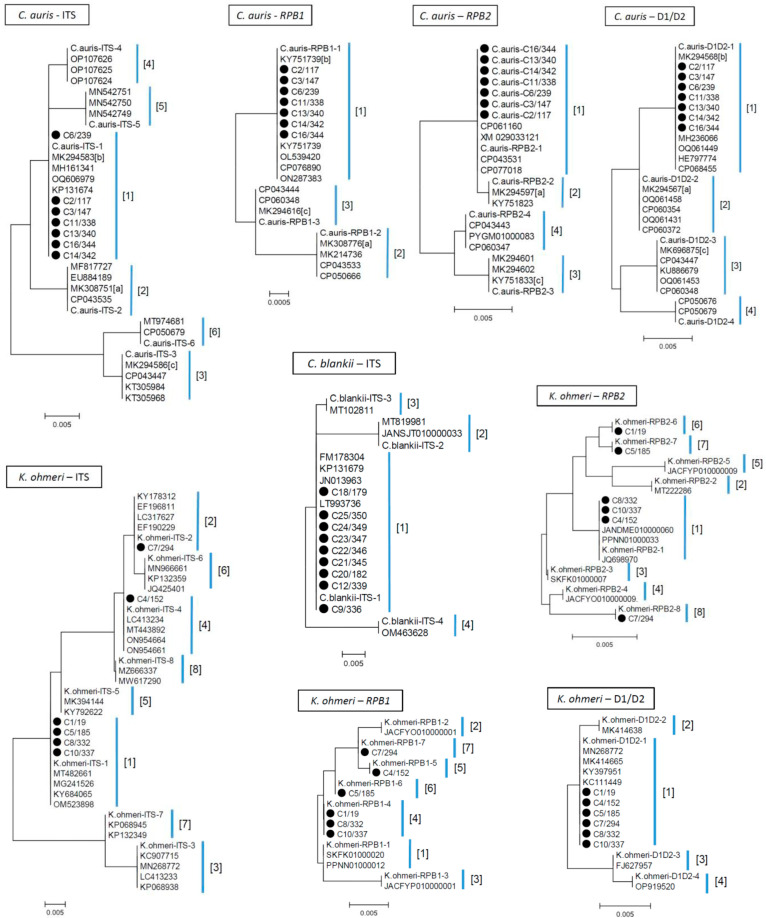
Phylogenetic dendrograms of four MLST loci of *C. auris* and *K. ohmeri*, and ITS locus of *C. blankii*, constructed using the maximum likelihood method with the MEGA.6 program. Sequences determined in the present study for Bangladeshi isolates are marked with a filled circle. Other sequences include those of individual locus types defined in Table 2, and those representing each locus type retrieved from GenBank database are indicated with accession numbers. Trees were statistically supported by bootstrapping with 1000 replicates, and genetic distances were calculated using the Kimura two-parameter model. The variation scale is presented at the bottom. Locus type numbers are shown in brackets on the right of the dendrogram with vertical bars.

**Table 1 jof-09-00658-t001:** Primers used for MLST of *C. auris*, *C. blankii*, and *K. ohmeri*, and mutation analysis of *C. auris ERG11* gene.

Species	Target	Primer Name	Sequence (5′-3′) [Orientation]	Reference
*C. auris, C. blankii, K. ohmeri*	ITS region, ribosomal subunit	ITS-1	TCCGTAGGTGAACCTTGCGG [+]	[24]
		ITS-4	TCCTCCGCTTATTGATATGC [-]	
	D1/D2, large ribosomal subunit region	NL-1	GCATATCAATAAGCGGAGGAAAAG [+]	[24]
		NL-4	GGTCCGTGTTTCAAGACGG [-]	
	*RPB1*, largest subunit of RNA polymerase II	RPB1af	GARTGYCCDGGDCAYTTYGG [+]	[24]
		RPB1Cr	CCNGCDATNTCRTTRTCCATRTA [-]	
*C. auris, K. ohmeri*	*RPB2*, DNA-dependent RNA polymerase II subunit	RPB2-5Fa	GACGATAGAGATCACTTTGG [+]	This study
		RPB2-7Cra	CCCATAGCTTGCTTACCCAT [-]	
*C. blankii*		CanBk-RPB2-F1	TCGTATGCTTCTAGTGGCTC [+]	
		CanBk-RPB2-R1	GGACATGGTATCCATACGAAC [-]	
*C. auris*	*ERG11*	CauErg11F	GTGCCCATCGTCTACAACCT [+]	[27]
		CauErg11R	TCTCCCACTCGATTTCTGCT [-]	

**Table 2 jof-09-00658-t002:** Assignment of allele type numbers (four MLST loci) of *C. auris*, *C. blankii*, and *K. ohmeri* and frequency of allelic sequences in GenBank database.

*C. auris* Locus (Nucleotide Length)											
ITS (376–379 nt.)			*RPB1* (621nt.)			*RPB2* (1014 nt.)			D1/D2 (536–540 nt.)		
type *^1^	GenBank accession no.*^2^	Frequency *^3^	type *^1^	GenBank accession no. *^2^	Frequency *^3^	type *^1^	GenBank accession no. *^2^	Frequency *^3^	type *^1^	GenBank accession no. *^2^	Frequency *^3^
1 [b]	MK294583	55	1 [b]	KY751739	39	1	CP043531	40	1 [b]	MK294568	55
2 [a]	MK308751	12	2 [a]	MK308776	6	2 [a]	MK294597	2	2 [a]	MK294567	35
3 [c]	MK294586	7	3 [c]	MK294616	3	3 [c]	KY751833	3	3 [c]	MK696875	22
4	OP107623	8				4	CP043443	3	4	CP050676	2
5	MN542749	4									
6	MT974681	2									
*C. blankii* locus (Nucleotide Length)											
ITS (445–446 nt.)			*RPB1* (633 nt.)			*RPB2* (1004 nt.)			D1/D2 (582 nt.)		
type	GenBank accession no.	Frequency	type	GenBank accession no.	Frequency	type	GenBank accession no.	Frequency	type	GenBank accession no.	Frequency
1	LT993736	10	1	EU344090	1	1	JQ699004	1	1	MF940140	3
2	MT819981	2				2	OQ603332 †	1	2	KY106326	1
3	MT102811	1									
4	OM463628	1									
*K. ohmeri* Locus (Nucleotide Length)											
ITS (382–385 nt.)			*RPB1* *^4^ (674 nt.)			*RPB2* *^4^ (989 nt.)			D1/D2 (526 nt.)		
type	GenBank accession no.	Frequency	type	GenBank accession no.	Frequency	type	GenBank accession no.	Frequency	type	GenBank accession no.	Frequency
1	MT482661	15	1	SKFK01000020	4	1	JQ698970	4	1	MN268772	29
2	EF190229	14	2	JACFYO010000001	1	2	MT222286	1	2	MK414638	2
3	KC907715	13	3	JACFYP010000001	1	3	SKFK01000007	1	3	FJ627957	2
4	LC413234	12	4	OQ603326 †	1	4	JACFYO010000009	1	4	OP919520	2
5	MK394144	6	5	OQ603327 †	1	5	JACFYP010000009	1			
6	MN966661	4	6	OQ603328 †	1	6	OQ603333 †	1			
7	KP132349	3	7	OQ603329 †	1	7	OQ603335 †	1			
8	MW617290	2				8	OQ603336 †	1			

*^1^ Allele ID is shown as Arabic numerals. Parenthesis indicates allele designation descrived by Kwon et al. [25]. Allelic sequence identified in only one accession number in GenBank database (frequency, one) was omitted for all the loci of *C. auris*, and ITS and D1/D2 loci of *K. ohmeri*. Sequences of individual loci of the three species are show in Appendix A. *^2^ One representative accession number is shown. † Accession no. obtained in the present study (frequency is shown as one). *^3^ Frequency indicates total number of the identical sequence with the same nucleotide length in GenBank database that was explored by BLAST search (as of March 2023). *^4^ Region of sequence from whole genome database: SKFK01000020, complement (41023..41696); JACFYO010000001, complement (17147..17820); JACFYP010000001, 2121625..212229; 8SKFK01000007, complement (1573048..1574036); JACFYO010000009, complement (422694..423682); JACFYP010000009, complement (422420..423408).

**Table 3 jof-09-00658-t003:** STs identified for *C. auris*, *C. blankii* and *K. ohmeri* isolates in this study.

Species	Allelic Profile				ST	No. of Isolates
	ITS	*RPB1*	*RPB2*	D1/D2		
*C. auris*	1	1	1	1	5 *	7
*C. blankii*	1	1	2	1	1	9
*K. ohmeri*	1	4	1	1	1	2
	1	4	6	1	2	1
	1	6	7	1	3	1
	2	7	8	1	4	1
	4	5	1	1	5	1

* Kwon et al. [25] described *C. auris* STs (ST clusters) 1-4 for following allelic profiles (ITS-*RPB1*-*RPB2*-D1/D2): ST1, 1-1-2-1 (b-b-a-b); ST2, 2-2-2-2 (a-a-a-a); ST3, 1-1-2-2 (b-b-a-a); ST4, 3-3-3-3 (c-c-c-c). ST5 was assigned in the present study.

**Table 4 jof-09-00658-t004:** Genotype and antifungal susceptibility, and patient information of *C. auris*, *C. blankii*, and *K. ohmeri* isolates.

Species	Isolate ID	ST	*C. auris* clade	Fluconazole		Amphotericin B		Patient				
				MIC (μg/mL)	Susceptibility *^1^	MIC (μg/mL)	Susceptibility *^1^	Age/Sex	Specimen	Symptom	Risk Factor *^2^	Outcome
*C. auris*	C2/117	5	I	1	S	0.5	S	6 day/F	blood	septicaemia	PT, LBW, PRAB, IVC	revovered
	C3/147	5	I	64	R	2	R	15 day/M	blood	septicaemia	PT, LBW, PRAB, IVC, TPN, PROM	deceased
	C6/239	5	I	32	R	2	R	30 day/M	blood	septicaemia	PT, LBW, PRAB, IVC	deceased
	C11/338	5	I	64	R	4	R	2 day/M	blood	septicaemia	HB	deceased
	C13/340	5	I	32	R	4	R	1 day/M	blood	septicaemia	PT, LBW	deceased
	C14/342	5	I	32	R	4	R	22 day/M	blood	septicaemia	PRAB, IVC	deceased
	C16/344	5	I	16	S	4	R	2 day/M	blood	septicaemia	LBW, PT	deceased
*C. blankii*	C9/336	1		64	N/A	4	N/A	3 day/M	blood	septicaemia	HB	deceased
	C12/339	1		64	N/A	4	N/A	3 day/F	blood	septicaemia	PT, LBW	deceased
	C18/179	1		2	N/A	4	N/A	14 y/F	aural swab	otomycosis	PRAB	recurrence
	C20/182	1		2	N/A	1	N/A	10 day/M	blood	septicaemia	LBW, PRAB, IVC	recovered
	C21/345	1		2	N/A	1	N/A	12 day/M	blood	septicaemia	PRAB, IVC	recovered
	C22/346	1		2	N/A	1	N/A	13 day/M	blood	septicaemia	PRAB, IVC	NI *^3^
	C23/347	1		2	N/A	1	N/A	6 day/F	blood	septicaemia	LSCS, PT, PRAB, IVC	recovered
	C24/349	1		2	N/A	1	N/A	6 day/F	blood	septicaemia	HB, PT, LBW, PROM	deceased
	C25/350	1		2	N/A	1	N/A	14 day/M	blood	septicaemia	HB, PT, LBW	deceased
*K. ohmeri*	C8/332	1		2	N/A	0.25	N/A	10 month/F	blood	septicaemia	PRAB	recovered
	C10/337	1		2	N/A	0.25	N/A	1 day/M	blood	septicaemia		recovered
	C1/19	2		64	N/A	0.25	N/A	40 y/F	aural swab	otomycosis	DM	recurrence
	C5/185	3		64	N/A	2	N/A	42 y/F	aural swab	otomycosis	CSOM	recurrence
	C7/294	4		2	N/A	0.25	N/A	10 day/M	blood	septicaemia		recovered
	C4/152	5		64	N/A	0.25	N/A	60 y/F	aural swab	otomycosis	CSOM, DM	recurrence

*^1^ Abbreviations: S, susceptible; R, resistant; N/A, not applicable. *^2^ Abbreviations: CSOM, chronic suppurative otitis media; DM, diabetes mellitus; HB, home birth; IVC, intravenous canula; LBW, low birth weight; LSCS, lower segment caesarean section; PRAB, prolonged antibiotic use (>7 days); PROM, premature rupture of membrane; PT, preterm; TPN, total parenteral nutrition. *^3^ No information was available.

## Data Availability

Data is contained within the article or Appendix A.

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
