# Peer review of "Clonal Diversity of Candida auris, Candida blankii, and Kodamaea ohmeri Isolated from Septicemia and Otomycosis in Bangladesh as Determined by Multilocus Sequence Typing"

_jof, 2023, doi:10.3390/jof9060658_

Round 1

Reviewer 1 Report

Sathi and colleagues evaluated the intraspecific diversity of clinical isolates of the emerging pathogens C. auris, C. blankii, and Kodamaea ohmeri by using the MLST typing method. The results are relevant to the field and also contribute to a better understanding of the potential heterogeneity among K. ohmeri. I have some comments:

-       All manuscript:

Typing errors, italicizing genera and species, and English grammar should be revised throughout the text and tables. Some examples:

Line 110: reciovered

Table 4: revovered

Lines 247, 250, 253: species name non italicized

-       Introduction:

The introduction is quite long and does not deeply address the relevance of typing the species used; a topic that will improve the manuscript. Candida glabrata, C. krusei, and C. guilliermondii are now renamed out from the genus Candida, and the new nomenclature should be used if the authors will keep using these species in the manuscript [e.g., Meyerozyma guilliermondii (syn. C. guilliermondii)].

Line 74 “Kodamaea ohmeri, which belongs to the Saccharomycetaceae family, is a teleomorph of C. guilliermondii var. membranaefaciens”: I suggest the authors rewrite the sentence once C. guilliermondii var. membranaefaciens is mentioned as a heterotypic synonym of K. ohmeri. In addition, as mentioned above, C. guilliermondii was renamed M. guilliermondii, and that variety is not a valid name.

-       Material and Methods

Lines 110-111: I suggest the authors go straight to the number and period of isolates studied than mentioning that the isolates were analyzed in previous studies. The authors used two sets of isolates: set 1 (n=109) used in a previous study and set 2 (n=22), included in this study. Perhaps it will be easier for the reader to understand if the authors combine the two sets into a single one.

Lines 123-125: If the isolates were accurately identified by rDNA sequencing the phenotypic methods are nonnecessary unless correlations between methods were evaluated. In addition, the “ITS1-5.8S-ITS2 region” may be replaced by the ITS region (over the manuscript).

Line 130: What values of e-value, percentage of identity, and coverage were considered during the Blast analysis? 

For further studies, I suggest to the authors use V9G and LS256 primer pairs to PCR and sequence the ITS region to avoid loss of nucleotides during the assembly in rare and emerging yeasts.

Line 135 (all topic): The use of disk diffusion for species where there are no guidelines for reading interpretation (such as those used) may not be able to solve problems and does not contribute to peculiarities in the sensitivity profile of the studied agents. It is not indicated to use the metric of other species (e.g., C. albicans) for the evaluation of C. auris, C. blankii, and K. ohmeri. The full topic needs to be revised and adjusted.

We already have tentative MIC Breakpoints for C. auris. Please check the values and adjust them in the manuscript (https://www.cdc.gov/fungal/candida-auris/c-auris-antifungal.html).

Why do the authors use MLST as the typing method instead of STR units? I suppose may be related to the availability of fragments analysis in their centers, but this should be included as a limitation of the study once microsatellites have shown better results than MLST for typing C. auris clinical isolates.

Line 148: the original MLST scheme for C. auris was proposed by Candejas-bueno et. (J Clin Microbiol. 2012 Nov; 50(11): 3641–3651). Please, check and correct.

Line 179 “of prototype C. auris strain B11220”: is this isolate the wild-type strain? If yes, this should be mentioned.

Line 180: Do the CladID strategy also checked by using the ITS-typing model?

-       Results

This section may be better adjusted to make it easy to read.

It was hard to understand when the 109 isolates were used and where it results is. Initially, the authors mentioned 20 isolates obtained from C. auris, C. blankii, and K. ohmeri that suddenly changes to 22.

I am not sure if “validation” (line 199) is the best word used once no one extension in the original panel was performed

Already included in comments and suggestions.

Author Response

Comment 1: Sathi and colleagues evaluated the intraspecific diversity of clinical isolates of the emerging pathogens C. auris, C. blankii, and Kodamaea ohmeri by using the MLST typing method. The results are relevant to the field and also contribute to a better understanding of the potential heterogeneity among K. ohmeri. I have some comments:

-All manuscript: Typing errors, italicizing genera and species, and English grammar should be revised throughout the text and tables. Some examples: Line 110: reciovered  Table 4: revovered  Lines 247, 250, 253: species name non italicized

Response: Thank you for pointing them out. In the revision process, we checked whole manuscript carefully and corrected errors of spelling, italic, and also English grammar. 

Comment 2: - Introduction: The introduction is quite long and does not deeply address the relevance of typing the species used; a topic that will improve the manuscript. Candida glabrata, C. krusei, and C. guilliermondii are now renamed out from the genus Candida, and the new nomenclature should be used if the authors will keep using these species in the manuscript [e.g., Meyerozyma guilliermondii (syn. C. guilliermondii)].

Response: Thank you for the suggestion. According to the comment, Introduction was shortened by deleting portions not related the main research topic of this manuscript. In this process, descriptions including “Candida glabrata, C. krusei” were deleted.

Comment 3: Line 74 “Kodamaea ohmeri, which belongs to the Saccharomycetaceae family, is a teleomorph of C. guilliermondii var. membranaefaciens”: I suggest the authors rewrite the sentence once C. guilliermondii var. membranaefaciens is mentioned as a heterotypic synonym of K. ohmeri. In addition, as mentioned above, C. guilliermondii was renamed M. guilliermondii, and that variety is not a valid name.

Response: Thank you for the comments. This sentence was corrected as “Kodamaea ohmeri, a heterotypic synonym of C. guilliermondii var. membranaefaciens, is also regarded as an emerging fungal pathogen,……”.

Comment 4: -Material and Methods

Lines 110-111: I suggest the authors go straight to the number and period of isolates studied than mentioning that the isolates were analyzed in previous studies. The authors used two sets of isolates: set 1 (n=109) used in a previous study and set 2 (n=22), included in this study. Perhaps it will be easier for the reader to understand if the authors combine the two sets into a single one.

Response: Thank you for the comments. According to the recommendation, descriptions of our previous study (Sathi et al., 2022) was shortened and directly wrote that 22 isolates were analyzed in this study.

Comment 5: Lines 123-125: If the isolates were accurately identified by rDNA sequencing the phenotypic methods are nonnecessary unless correlations between methods were evaluated. In addition, the “ITS1-5.8S-ITS2 region” may be replaced by the ITS region (over the manuscript).

Response: Thank you for the suggestion. In the revised manuscript, the part of phenotypic identifcation was deleted, and “ITS1-5.8S-ITS2 region” was corrected as “ITS region”.

Comment 6: Line 130: What values of e-value, percentage of identity, and coverage were considered during the Blast analysis? Id

Response: Thank you for checking it. We used e-value of 0.05, and identity of >97% and coverage of 100%. Identities of ITS locus of 3 species including values of our isolates are shown in Table S6, which support the accuracy of BLAST analysis. 

Comment 7: For further studies, I suggest to the authors use V9G and LS256 primer pairs to PCR and sequence the ITS region to avoid loss of nucleotides during the assembly in rare and emerging yeasts.

Response: Thank you for thr constructive suggestion. We will use these primers for further study in the near future.

Comment 8: Line 135 (all topic): The use of disk diffusion for species where there are no guidelines for reading interpretation (such as those used) may not be able to solve problems and does not contribute to peculiarities in the sensitivity profile of the studied agents. It is not indicated to use the metric of other species (e.g., C. albicans) for the evaluation of C. auris, C. blankii, and K. ohmeri. The full topic needs to be revised and adjusted.

Response: Thank you for pointing the important issue. I agree to the view mentioned. We measured MIC by broth microdillution test for only fluconazole, and amphotericin B. Therefore, in the revised manuscript, only these two antifungal agents were decribed. From the original manuscript, we deleted description of disk diffusion test (Methods section) and results of susceptibility test (Results section, Table 4). In the revised manuscript, only methods and results of broth microdillustion test with fluconazole and amphotericin B were written. Descriptions in Methods, Results and Table 4 were revised.

Comment 9: We already have tentative MIC Breakpoints for C. auris. Please check the values and adjust them in the manuscript (https://www.cdc.gov/fungal/candida-auris/c-auris-antifungal.html).

Response: Thank you for telling the breakpont for C. auris defined by CDC. In the revised manuscript, this criteria by CDC was added in Methods section, and judged susceptibility to fluconazole and amphotericin B according to that criteria. Results were shown in Table 4.

Comment 10: Why do the authors use MLST as the typing method instead of STR units? I suppose may be related to the availability of fragments analysis in their centers, but this should be included as a limitation of the study once microsatellites have shown better results than MLST for typing C. auris clinical isolates.

Response: Thank you for the comments. In this study, we focused on MLST for the three fungal species, that is the reason why we did not try STR typing for further analysis of C. auris. As the reviewer pointed, it is important to compare the resolution ability of typing between STR and MLST. In the revised manuscript, in the final paragraph of Discussion, STR typing for C. auris was briefly written, refering to a limitation in our present study.  

Comment 11: Line 148: the original MLST scheme for C. auris was proposed by Candejas-bueno et. (J Clin Microbiol. 2012 Nov; 50(11): 3641–3651). Please, check and correct.

Response: Thank you for the suggestion. In the revised version, literature by Cendejas-Bueno et al. 2012 was also cited as the original MLST scheme of C. auris.

Comment 12: Line 179 “of prototype C. auris strain B11220”: is this isolate the wild-type strain? If yes, this should be mentioned.

Response: Thank you for the comment. When we checked again, we found that B11220 was early isolate of C. auris (2009, in Japan) but it is not the name of prototype strain. This is a wil type strain. Therefore, the description of this strain was corrected as “fluconazole-susceptible C. auris strain B11220 (wild type strain of clade II, GenBank accession no. XP_028891800) [28].” with a reference of this strain.

Comment 13 : Line 180: Do the CladID strategy also checked by using the ITS-typing model?

Response: CladID scheme published by Narayanan (Microbiol. Spectr. 2022) is PCR-based typing. Target genes are not ITS, but this scheme was dsigned from comparison of whole genomes to detect Clade-specific region and its accuracy in identification is established.

Comment 14: -Results

This section may be better adjusted to make it easy to read.

It was hard to understand when the 109 isolates were used and where it results is. Initially, the authors mentioned 20 isolates obtained from C. auris, C. blankii, and K. ohmeri that suddenly changes to 22.

Response: Thank you for the comments. Description of the 109 isolates in the previous study was deleted. Therefore, in the revised version, readers may easily understand our present study of 22 isolates.

Comment 15: I am not sure if “validation” (line 199) is the best word used once no one extension in the original panel was performed

Response: Thank you for pointing it out. The word “validation” was deleted, and subtitle was corrected as “MLST scheme of C. auris”.

Reviewer 2 Report

I would recommend to perform phylogenetic analysis on each gene for all isolates in this study as well as the ones available in Genbank. This type of visualization would be more informative than just reporting MLSTs in tables.

The discussion is too extensive based on the data presented and needs to be shortened.

Methods:

Because it was previously published, lines 108-122 should be shortened or simply referenced to the previous publication. 2.3. and 2.4. should be combined as the majority of methods used are the same.

Results:

It is not clear why the letter designations for MLST were converted to Arabic numerals. It should be kept consistent between different studies and different labs.

Why are the MLSTs for C. blankii and K. ohmeri called ‘provisional’?

Table 3: I would remove the allelic profiles that were not detected in C. auris.

For 3.5. it would be important to perform statistical analysis to test if there is indeed association/correlation of resistant or susceptible isolates with specific patient groups.

Finally, why not obtain complete gene sequences for the MLST loci used?

Line 48: should be ‘non-albicans

Line 62: should be ‘multi-drug resistance (MDR)’

The manuscript will need moderate language editing

Author Response

Comment 1: I would recommend to perform phylogenetic analysis on each gene for all isolates in this study as well as the ones available in Genbank. This type of visualization would be more informative than just reporting MLSTs in tables.

Response: Thank you for the suggestion. According to this suggestion, we created phylogenetic dendrograms of 4 loci of C. auris and K. ohmeri, and ITS locus of C. blankii, using sequences of our present isolates and those from GenBank. These were compiled and shown as Figure 1, in the revised manuscript. This analysis of RPB1, RPB2, and D1/D2 of C. blankii was omitted, because locus types were too less.

Comment 2: The discussion is too extensive based on the data presented and needs to be shortened.

Response: Thank you for the comment. Discussion was substantially shortened.

Comment 3: Methods: Because it was previously published, lines 108-122 should be shortened or simply referenced to the previous publication. 2.3. and 2.4. should be combined as the majority of methods used are the same.

Response: Thank you for the comment. Methods on isolation of Candida was shortened, adding a refrence. Sections 2.3 and 2.4 were combined, to avoid repetition of similar descriptions.

Comment 4: Results: It is not clear why the letter designations for MLST were converted to Arabic numerals. It should be kept consistent between different studies and different labs.

Response: Thank you for the comment. I agree to your view. However, MLST of all the bacteria and fungi are shown as combination of arabic numerals. In this regard, combination of alphabet shown by Kwon et al. (e.g., a-a-a-a, b-b-a-b, etc.) is uncommon description. Therefore, in the revised manuscript, the reason why the alphabet was coverted to number was written (line 169-170). In addition, in the footnote of Table 3, original MLST descriptions by alphabet were also shown with our present description of allelic profiles.

Comment 5: Why are the MLSTs for C. blankii and K. ohmeri called ‘provisional’?

Response: In original version, “provisional” was added because we attempted these MLST first time for C. blankii and K. ohmeri, and may be modified by other researchers in the future. However, this is not really original scheme but already designed and applied to some Candida species (Cendejas-Bueno et al., J Clin Microbiol. 2012, 50:3641), therefore, “provisional” was deleted in the revised manuscript. 

Comment 6 : Table 3: I would remove the allelic profiles that were not detected in C. auris.

Response: Thank you for the suggestion. Table 3 was revised, to delete allelic profiles that were not detected for present isolates. Those allelic profiles were shown in footnote.

Comment 7: For 3.5. it would be important to perform statistical analysis to test if there is indeed association/correlation of resistant or susceptible isolates with specific patient groups.

Response: Thank you for the suggestion. I agree to your view. However, in this study, because numbers of isolates were low, it is difficult to determine correlation of patients group and traits of isolates. Therefore, in this study, we just described the findings as they are as shown in Table 4, without telling any correlation among factors. 

Comment 8: Finally, why not obtain complete gene sequences for the MLST loci used?

Response: Thank you for the question. For MLSTscheme of bacteria and fungus, partial sequences which is a fixed range (c.a. 400-1000bp) of housekeeping genes specifically defined for each microorganism, are always used. The reason why full-length gene is not used is not clear, but I think that the scheme was designed for practical convenience, by using PCR and direct sequencing of PCR products.

Comment 9: Line 48: should be ‘non-albicans’ Line 62: should be ‘multi-drug resistance (MDR)’

Response : Thank you for checking these words. These were corrected. The description of “non-albicans” was deleted to shorten Introduction section.

Round 2

Reviewer 1 Report

Authors provided an substantial improvements i the manuscript, also answering all the reviewer´s comments.

Topics were addressed.

Revised.

Reviewer 2 Report

The authors have made all necessary revisions.

The authors have made all necessary revisions.